# Mediterranean Diet Pyramid: A Proposal for Italian People. A Systematic Review of Prospective Studies to Derive Serving Sizes

**DOI:** 10.3390/nu11061296

**Published:** 2019-06-07

**Authors:** Annunziata D’Alessandro, Luisa Lampignano, Giovanni De Pergola

**Affiliations:** 1Medical Endocrinologist, General Internal Medicine A.S.L. Bari, v.le Iapigia 38/g, 70126 Bari, Italy; 2National Institute of Gastroenterology “S. de Bellis”, Research Hospital, Castellana Grotte, 70013 Bari, Italy; luisalampignano@gmail.com; 3Department of Biomedical Sciences and Human Oncology, Section of Internal Medicine and Oncology, School of Medicine, Policlinico, University of Bari “Aldo Moro”, p.zza Giulio Cesare 11, 70124 Bari, Italy; gdepergola@libero.it

**Keywords:** Mediterranean diet, Mediterranean diet pyramid, noncommunicable diseases

## Abstract

In the last decade, a number of meta-analyses of mostly observational studies evaluated the relation between the intake of food groups and the risk of noncommunicable diseases (NCDs). In this study, we systematically reviewed dose-response meta-analyses of prospective studies with the aim to derive the quantities of food to consume to attain a protective (Mediterranean food) or a non-adverse (non-Mediterranean food) effect toward selected NCDs such as cardiovascular disease (CVD) including coronary heart disease (CHD) and stroke, type 2 diabetes (T2DM), colorectal (CRC) and breast cancer. These derived quantities, wherever possible, were suggested for a quantification of food servings of the Mediterranean Diet Pyramid proposed for Italian People (MDPPI). This pyramid came from the Modern Mediterranean Diet Pyramid developed in 2009 for Italian people. A weekly menu plan was built on the advice about frequency of intakes and serving sizes of such pyramid and the nutritional composition of this diet was compared with the Reference Italian Mediterranean Diet followed in 1960 in Nicotera. The diet built according the advice of MDPPI was very similar to that of Nicotera in the late 1950s that has been chosen as Italian Reference Mediterranean Diet with the exception of percentage of energy provided by cereals that was lower and of fruits and vegetables that was higher. Saturated fatty acids were only the 6% of daily energy intake. Also the Mediterranean Adequacy Index (MAI) was very similar to that of the aforementioned diet.

## 1. Introduction

According to the 2018 Global Health Observatory Data of the World Health Organization (WHO), noncommunicable diseases (NCDs) such as cardiovascular disease (CVD), type 2 diabetes (T2DM), cancer and chronic respiratory diseases represent the leading cause of death in the world [1]. In Italy 91% of deaths are caused by NCDs and 10% of them are premature deaths because it affects people in a 30 to 70-year-old span [2].

Some lifestyle modifications and among them the adoption of healthy dietary choices as well as increasing the intake of fresh fruit and vegetable, whole grains and healthy fats, represent useful measures for the prevention of NCDs [3].

The Dietary Guidelines Advisory Committee included the Mediterranean Diet among highly beneficial dietary patterns for the prevention of overweight and obesity, CVD, T2DM, CRC and post-menopause breast cancer based upon prospective cohort studies, randomized clinical trials (RCTs) and high-quality systematic reviews [4].

The definition of dietary pattern takes into account several factors linked to habitually consumed food and beverages: quantities, proportions, combination or variety and frequency of intake [5].

The Mediterranean Diet is a dietary pattern that was identified in the early 1960s in South Italy, Crete and other areas of Greece [6]. At that time the food intake habits of three cohorts of the Seven Countries Study, Corfu and Crete in Greece and Nicotera in South Italy were almost identical [7]. The dietary habits characterized by higher consumption of vegetables and lower consumption of animal food were considered important determinants of the very low mortality for CHD observed in the Corfu and Crete cohorts at 25-year of follow-up [8].

During the last decade, a number of meta-analyses of mostly observational studies evaluated the association between the intake of food groups and the development of NCDs.

In the present study we performed a systematic review of dose-response meta-analyses of prospective studies, which evaluated the association between the intake of food groups belonging to a variant of the Modern Mediterranean Diet Pyramid (www.inran.it) and the risk of CVD, including CHD and stroke, T2DM, CRC and breast cancer. This variant was the MDPPI [9] (Figure 1).

In particular, we derived from these meta-analyses the serving sizes of food to be consumed in order to obtain a protective (Mediterranean food) or a not detrimental (non-Mediterranean food) effect toward selected NCDs. A weekly menu plan was built on the advice about frequency of intakes and serving sizes of MDPPI and the nutritional composition of this diet was compared with the Reference Italian Mediterranean Diet followed in 1960 in Nicotera [10].

## 2. First Section

### 2.1. Methods

The criteria for systematic reviews (PRISMA statement) were followed [11].

We searched on Medline and Google Scholar for dose-response meta-analyses investigating the association between food groups of the MDPPI such as whole grains, vegetables, fruits, milk, cheese, yogurt, nuts, olive oil, herbs, spices, fish, legumes, eggs, refined grains, sweets/cakes/cookies, potatoes, red meat, processed meat, poultry, and red wine, on CVD, CHD, stroke, T2DM, CRC, and breast cancer risk, up to December 2018.

Inclusion criteria were as follows: (1) linear and/or nonlinear dose-response meta-analyses of prospective studies (cohort studies, follow-up of RCTs, case-cohort studies, nested case-control studies); (2) summary relative risks (RRs) or summary hazard ratios (HRs) with 95% confidence intervals (CIs); (3) I^2^ statistic or *p*-value for heterogeneity; (4) exposure expressed in metric units (e.g., grams per day, milliliters per day; unit per day or per week, for eggs).

The publications were selected if the word meta-analysis appeared in the title and/or in the abstract. They were assessed for eligibility after reading the full text.

### 2.2. Results

Out of 95 dose-response meta-analyses that were identified, 36 were excluded while 59 met inclusion criteria and were withheld for this systematic review.

Twelve dose-response meta-analyses were evaluated for whole grain intake [12,13,14,15,16,17,18,19,20,21,22,23]; 11 for vegetable intake [17,18,20,21,24,25,26,27,28,29,30]; 12 for fruit intake [17,18,20,21,24,25,26,27,28,29,30,31]; 13 for milk intake [20,32,33,34,35,36,37,38,39,40,41,42,43]; 8 for cheese intake [20,33,35,36,37,38,39,44]; 6 for yogurt intake [33,35,36,37,38,41]; 8 for nut intake [17,18,21,41,45,46,47,48]; 2 for olive oil intake [49,50]; 9 for fish intake [17,18,20,21,41,51,52,53,54]; 4 for legume intake [17,18,20,21]; 3 for egg intake [55,56,57]; 2 for refined grain intake [17,18]; 1 for potato intake [58]; 13 for red meat intake [17,18,20,21,41,59,60,61,62,63,64,65,66]; 13 for processed meat intake [17,18,20,21,41,59,60,61,62,64,65,66,67]; 3 for poultry intake [41,61,68]; 2 for red wine intake [69,70].

No dose-response meta-analyses were identified for spice, herb and sweet/cake/cookie intakes.

#### 2.2.1. Whole Grains

Table 1 reports the summary of linear dose-response meta-analyses of prospective studies on whole grain intake and CVD, CHD, stroke, type 2 diabetes (T2DM), and CRC.

A meta-analysis reported both a dose-response analysis for specific whole grains and for total whole grains [12]. Each increment intake of 90 g/day reduced CVD risk by 22% [16], CHD risk by 19% [16], and CRC risk by 17% [19,20]. Each increment intake of 90 g/day of specific whole grains reduced CVD mortality by 17% [12]. A meta-analysis reported that each increment intake of 30 g/day of whole grains reduced CHD risk by 5% [17]. In these meta-analyses, the heterogeneity was low or moderate (I^2^ statistic < 50%).

Nonlinear dose-response analyses. The dose-response analysis of the association between whole grain intake and CVD mortality was nonlinear and above an intake of ~35 g/day [13] or of ~60 g/day [22] additional but more moderate benefits were evident. Two dose-response meta-analyses reported no evidence of nonlinearity between whole grain intake and reduction of CVD mortality [12,23].

There was evidence of a nonlinear association between whole grain intake and CVD, CHD and stroke with a reduction in risk up to 200 g/day for CVD, 210 g/day for CHD, and 120–150 g/day for stroke [16] and no reductions above these intakes. However, Bechthold et al., found evidence of a nonlinear dose-response association between whole grain intake and CHD with a decreased risk of 17% with increasing intake up to ~100 g/day and no benefits with further intakes [17]. In a meta-analysis, no association was evident for whole grain intake and stroke risk in the nonlinear dose-response analysis [17].

There was evidence of a nonlinear dose-response association between whole grain intake and T2DM risk with a 25% of risk reduction along with an increasing intake up to ~50 g/day and minimal benefits above this level of intake [18].

The intake of whole grains was associated with a linear decrease in the risk of CRC [19,21]; the risk decreased by ~20% along an increasing intake up to ~120 g/day and further benefits were evident for higher intakes up to ~200 g/day [21].

#### 2.2.2. Vegetables

Table 2 reports the summary of linear dose-response meta-analyses of prospective studies on vegetable intake and CVD, CHD, stroke, T2DM, CRC, and breast cancer.

Each daily increment intake of 200 g of vegetables reduced the risk of CVD by 10% [24], whereas a daily increment intake of 400 g reduced the CHD risk by 18% [25] and a daily increment intake of 100 g by 3% [17]. In these meta-analyses, the heterogeneity was below 50%.

Nonlinear dose-response analyses. There was evidence of nonlinearity for vegetable intake and CVD (although the association was almost linear) with a 28% risk reduction at intakes of 600 g/day [24]. The nonlinear dose-response analysis was significant for the association between vegetable intake and CVD risk with a reduction of 11% and 28% for a daily intake of 200 g and 600 g, respectively [29].

A nonlinear association of CHD risk with vegetable intake was evident [24,25] with a reduction in risk at the lower levels of intake up to ~200–300 g/day and light further reductions in RRs of 30% up to 550–600 g/day [24]. One dose-response meta-analysis reported that the association between vegetable intake and CHD was linear; an increasing intake of vegetables up to 400 g/day reduced the RRs of CHD by ~12% and higher intakes lead to further reductions in RRs [17]. 

For stroke risk, there was no evidence of nonlinearity: risk reduction of stroke was evident along with the entire range of vegetable intake with the strongest reductions in RRs of 20% for a daily intake of 400 g [30]. However, in two dose-response meta-analyses there was evidence of nonlinearity between vegetable intake and stroke risk with a 28% reduction in RRs at 500 g/day [24] or with ~12% reduction in RRs up to intakes of ~200 g/day [17] without additional benefits above these amounts.

The association between vegetable intake and T2DM risk was nonlinear [18,26,27]. A maximal reduction in RRs of 9% was evident at ~300 g/day of intakes with no additional benefits for higher intakes [18]. Similarly, in another meta-analysis, the T2DM risk decreased by 6% at intakes of ~200–300 g/day of vegetables without further benefits at higher intakes [27].

There was evidence of nonlinear dose-response between vegetable intake and CRC risk: the risk decreased along the entire range of intakes and the RR reduction were 7% up to ~200 g/day; minimal further reductions were evident for higher intakes [21].

There was no evidence of nonlinear dose-response association between vegetable intakes and breast cancer risk [28].

#### 2.2.3. Fruits

Table 3 reports the summary of linear dose-response meta-analyses of prospective studies on fruit intake and CVD, CHD, stroke, T2DM, CRC, and breast cancer.

Each daily increment intake of 300 g or 200 g of fruits reduced the RRs of CHD by 16% [25] and 10% [24], respectively. The RRs of T2DM were reduced by 6%, 1% and 2% for a daily increment intake of 106 g [26], 106 g [27] and 100 g [18], respectively. The breast cancer risk was reduced by 6% for each daily increment intake of 200 g [28]. In these linear dose-response meta-analyses, the heterogeneity was below 50%.

Nonlinear dose-response analyses. There was evidence of nonlinearity between fruit intake and CVD risk with the strongest inverse association at lower intakes (up to ~200–300 g/day) and light further reductions at greater intakes: at 800 g/day of intake the reduction in RRs was 27% [24]. Similarly, in another meta-analysis, the reduction in CVD risk was 14% at 200 g/day of intake and 16% at 500 g/day of intake [29].

Nonlinear association between fruit intake and CHD was found: most of the reduction in risk was up to ~200 g/day of intake [17,24,25] without benefit for greater intake [17] or with a light reduction for greater intakes [24,25].

In two dose-response meta-analyses, the association between fruit intake and stroke was nonlinear with a 20% of reduction in RRs for daily intakes of ~200–350 g [24] or ~200 g [17] and not benefits for greater intakes. One dose-response meta-analysis reported no evidence of nonlinearity between fruit intake and stroke with a reduction in RRs of 46% for a daily intake of 300 g [30].

Four dose-response meta-analyses reported evidence of nonlinear association between fruit intake and T2DM risk with most reduction in RRs for lower intakes [18,26,27,31]. For ~200 g of fruit intake the reduction in RRs was 13% [31] 10% [18] or 12% [27] and no benefits were evident for higher intakes.

A nonlinear dose-response association was detected between fruit intake and CRC with the greatest reduction in RRs by ~8% increasing the intake up to ~200 g/day and little further benefits above these intakes [21]

The inverse association between fruit intake and breast cancer was linear [28].

#### 2.2.4. Milk

Table 4 reports the summary of linear dose-response meta-analyses of prospective studies on milk intake and CVD, CHD, stroke, T2DM, CRC, and breast cancer. 

A daily increment intake of 200 mL of milk reduced the RRs of CVD by 6% [32]. A daily increment intake of 200 mL [32,34] or 244 g of milk [33] had a neutral effect on CHD risk. The RRs of CRC was reduced by 10% [39] or 6% [20] for each daily increment intake of 200 g of milk. The RRs of breast cancer and breast cancer incidence were reduced by 2% [40] and 3% [41] for each daily increment intake of 200 g of milk. In these meta-analyses, the heterogeneity was low below 50%.

Nonlinear dose-response analyses. An inverse and nonlinear association between milk intake and stroke was found with a maximal effect up to ~200 mL/day (reduction in RRs of 18%) [42] or up to 125 g/day (reduction in RRs of 14%) [35]; the reduction in RRs remained but was attenuated up to ~700 mL/day [42] or ~750 g/day [35].

There was evidence of an inverse linear association between low fat or skim milk intake and T2DM and of a nonlinear positive association between whole milk intake and T2DM [36].

The association between milk intake and reduced CRC risk was nonlinear with the strongest reduction in RRs (20–30%) from 500 to 800 g/day of intake and week association below 200 g/day of intake [39]. In a meta-analysis of 22 prospective cohort studies, no association between milk intake and breast cancer risk was found [43]. In another meta-analysis, a linear inverse association between skim milk intake and breast cancer incidence was found [41].

#### 2.2.5. Cheese

Table 5 reports the summary of linear dose-response meta-analyses of prospective studies on cheese intake and CVD, CHD, stroke, T2DM, and CRC. 

A daily increment intake of 50 g reduced CVD and CHD risk by 8% and 10%, respectively [44]. The stroke risk was reduced by 3% with a daily increment intake of 40 g [35]. An increment intake of 50 g reduced the T2DM risk by 8% [36].

The risk of CRC was reduced by 4% [39] and 6% [20] with a daily increment intake of 50 g, respectively. In these dose-response meta-analyses, the heterogeneity was below 50%.

Nonlinear dose-response analyses. The association between cheese intake and CVD was nonlinear and almost U-shaped with a maximal risk reduction at ~40 g/day [44]. The association between cheese intake and CHD was linear with progressive risk reduction up to an intake of 120 g/day [44]. The association between cheese intake and stroke was inverse and nonlinear with a maximal risk reduction at 40 g/day [44] or at 25 g/day [35] without benefits for greater intakes.

Nonlinearity was found between cheese intake and T2DM with a maximal reduction in RRs up to 50 g/day and not further benefits for higher intakes [36].

#### 2.2.6. Yogurt

Table 6 reports the summary of linear dose-response meta-analyses of prospective studies on yogurt intake and CVD, CHD, stroke, T2DM, and breast cancer. 

A daily increment intake of 50 g [33] or 100 g [35] was not associated with CVD, CHD [33] and stroke risk [35]. A daily increment intake of 200 g reduced breast cancer incidence by 13% [41]. In these dose-response meta-analyses, the heterogeneity was below 50%.

Nonlinear dose-response analyses. The association between yogurt intake and T2DM risk was inverse and nonlinear with a maximal reduction at intakes of 120–140 g/day [36] or 80 g/day [38] and not further reductions at higher intakes. The association between yogurt intake and breast cancer risk was nonlinear [41].

#### 2.2.7. Nuts

Table 7 reports the summary of linear dose-response meta-analyses of prospective studies on nut intake and CVD, CHD, stroke, T2DM, CRC and breast cancer.

A daily increment of 28 g reduced the RRs of CVD incidence [45] and CHD [46] by 29%; the risk of stroke by 7% [46] or 1% [17]; the risk of CRC [18] and breast cancer incidence [41] by 4%. In these dose-response meta-analyses, the heterogeneity was below 50%.

Nonlinear dose-response analyses. The association between nut intake and CVD was nonlinear with a maximal reduction in risk up to intakes of ~15 g/day and no benefits above these values [46]. There was evidence of nonlinearity between nut intake and CHD risk with an inverse association that reached the maximal values at ~15–20 g of daily intakes without further reductions for greater intakes [46]. A nonlinear inverse association was reported between nut intake and CHD risk with maximal benefits in reduction of RRs (21%) up to ~10–15 g/day and no benefits above these intakes [17].

There was evidence of an inverse nonlinear association between nut intake and risk of stroke with a maximal reduction in RRs up to ~10–15 g of daily intake but a positive association at ~30 g of daily intake [46]. An inverse nonlinear association was found between nut intake and stroke risk with a 14% risk reduction at daily intake of 12 g without any further benefits for greater intakes [48]. However, in another meta-analysis no association was found between daily nut intake and risk of stroke in the nonlinear dose-response analysis [17]. There was no evidence of a nonlinear dose-response association between daily nut intake and T2DM [18] or CRC risk [21]. The association between nut intake and breast cancer was nonlinear [41].

#### 2.2.8. Olive Oil

Table 8 reports the summary of linear dose-response meta-analyses of prospective studies on olive oil intake and CHD, stroke, and T2DM.

Each 25 g increment intake per day of olive oil reduced the RRs of stroke by 24% [49]. An increment intake of 10 g of olive oil reduced the RRs of T2DM by 9% [50]. In these dose-response meta-analyses, the heterogeneity was below 50%.

Nonlinear dose-response analyses. There was evidence of an inverse nonlinear association between olive oil intake and T2DM risk. A daily intake of ~15–20 g reduced the RRs of T2DM by 13%. Greater intakes did not bring further benefits. Excluding one study the association became linear [50].

#### 2.2.9. Fish and Shellfish

Table 9 reports the summary of linear dose-response meta-analyses of fish intake and CVD, CHD, stroke, T2DM, CRC, breast cancer.

Each increment of 20 g/day decreased CVD mortality by 4% [51]. Each daily intake of 100 g reduced the RRs of CHD and stroke of 12% and 14%, respectively [17] and the CRC risk by 11% [20] or 7% [21]. A daily increment intake of 120 g increased the breast cancer incidence of 7% [41]. In these dose-response meta-analyses, the heterogeneity was below 50%.

Nonlinear dose-response analyses. The association between fish intake and CVD mortality was linear [51]. There was evidence of linearity of the association between fish intake and CHD mortality: for each 20g/day the reduction of RRs was 7% [54]. However, in another meta-analysis there was evidence of nonlinearity between fish intake and risk of CHD mortality that had greater risk reduction at ~30–60 g/day than for lower and higher intakes [52]. The nonlinear dose-response association between fish intake and CHD or stroke was significant and the risk of CHD decreased with greater daily intake up to ~15% at ~250 g/day; the risk of stroke decreased with greater daily intake up to ~10% at ~80–100 g/day [17].

No association between fish intake and T2DM [18], CRC [21], and breast cancer risk [53] was evident. A nonlinear association with fish intake and breast cancer incidence was found [41].

#### 2.2.10. Legumes

Table 10 reports the summary of linear dose-response meta-analyses of prospective studies on legume intake and CHD, stroke, T2DM, and CRC.

A daily increment of 50 g reduced the risk of CHD by 4% [17]. No association was evident for a daily increment of 50 g of legumes on the risk of CRC [20]. In these dose-response meta-analyses, the heterogeneity was below 50%.

Nonlinear dose-response analyses. An inverse nonlinear dose-response association between legume intake and CHD was evident; the risk decreased by ~10% with increasing intakes up to ~100 g/day without no benefits for higher intakes. No association was evident between legume intake and stroke [17], T2DM [18], and CRC risk [21].

#### 2.2.11. Eggs

Table 11 reports the summary of linear dose-response meta-analyses of prospective studies on egg intake and CHD, stroke, T2DM, and breast cancer.

An increased intake of 1 egg/day was not associated with increased risk of CHD or stroke [55]. An increased intake of 5 eggs/week increased the risk of breast cancer of 5% [57]. No heterogeneity was detected between the studies.

Nonlinear dose-response analyses. No evidence of nonlinear dose-response was found between egg intake and risk of CHD and stroke [55]. The association between egg intake and breast cancer risk was linear with an increased risk for intakes ≥5 eggs/week [57].

#### 2.2.12. Refined Grains

Table 12 reports the summary of linear dose-response meta-analyses of prospective studies on refined grain intake and CHD, stroke, T2DM.

Each additional daily intake of 30 g of refined grains was not associated with the risk of CHD or stroke without heterogeneity between the studies [17].

Nonlinear dose-response analyses. The association between refined grain intake and CHD was linear and the RRs were greater than 1.00 for intakes higher than ~100–120 g/day [17].

There was no association between refined grain intake and risk of stroke [17]. The association between refined grain intake and T2DM was linear and direct. An intake of ~200–400 g/day was associated with an increased risk of 6–14% [18].

#### 2.2.13. Potatoes

Table 13 reports the summary of linear dose-response meta-analyses of prospective studies on potato intake and CHD, stroke, T2DM, and CRC.

An increment in daily total potato intake of 150 g was not associated with the risk of CHD, stroke or CRC but was positively associated with T2DM risk (18% increase). No heterogeneity between the studies was found [58].

Nonlinear dose-response analyses. There was no association between total potato intake and CHD and stroke. There was no evidence of nonlinearity between total potato intake and T2DM or CRC risk. The risk of T2DM was positively associated with potato intake: at value up to ~260 g/day it reached ~51% mostly due to French fries intake [58].

The risk of CRC was positively associated with potato intake: for an intake higher than ~134 g/day, the risk the CRC increased up to 25% for a daily intake of ~190 g [58].

#### 2.2.14. Red Meat

Table 14 reports the summary of linear dose-response meta-analyses of prospective studies on red meat intake and CVD, CHD, stroke, T2DM, CRC, and breast cancer.

Each 100 g daily increment of red meat was associated with a 15% and a 12% increased risk of CHD and stroke, respectively [17]. The risk of T2DM was increased for the same daily intake by 13% [61]. The risk of CRC was increased by 28% for an increase in red meat intake of 120 g/day [62], and from 12% to 17% for an increase in red meat intake of 100 g/day [20,21,64,65]. A daily increment of 120 g of red meat intake increased the risk of breast cancer by 11% [66]. The heterogeneity between the studies was below 50%.

Nonlinear dose-response analyses. There was evidence of a nonlinear dose-response association between red meat intake and CHD risk: it increased by ~20% up to ~100 g of daily intake [17]. The association between red meat intake and stroke was linear with an increased risk of ~10% for increasing intakes up to ~100 g/day [17].

The association between read meat intake and T2DM risk was linear and direct with an increasing risk of ~20% for a daily intake up to ~100 g/day of intake [18]. The association between red meat intake and CRC risk was linear and positive and an intake of 150 g/day was associated with an increased risk of ~20% [21]. A linear association was observed between red meat intake and increased breast cancer incidence risk [41].

#### 2.2.15. Processed Meat

Table 15 reports the summary of linear dose-response meta-analyses of prospective studies on processed meat intake and CVD, CHD, stroke, T2DM, CRC, and breast cancer. 

Each daily increment of 50 g of processed meat intake increased CHD risk by 27% [17]. Each daily increment intake of 30 g increased CRC risk by 9% [62]. Each daily increment intake of 50 g increased CRC risk from 17% to 22% [20,21,64,65]. In two dose-response meta-analyses, the risk of breast cancer was increased by 9% with each increment of 50 g/day of processed meat [41,66]. The heterogeneity between the studies was below 50%.

Nonlinear dose-response analyses. There was no evidence of nonlinear dose-response association between processed meat and CHD or stroke risk. No association was found between processed meat intake and CHD risk while the risk of stroke increased by ~15% with a processed meat intake up to 70 g/day [17]. A nonlinear positive dose-response association was evident between processed meat intake and T2DM with a risk increase of ~30% up to 50 g/day and moderate further increase in risk for additional intakes [18]. The dose response association between processed meat intake and increased CRC risk was linear and an intake up to ~60 g/day increased the risk of ~20% [21]. The relation between processed meat intake and increased risk of breast cancer incidence was linear [41].

#### 2.2.16. Poultry

Table 16 reports the summary of linear dose-response meta-analyses of prospective studies on poultry intake and T2DM, CRC, and breast cancer.

Each daily 50 g increment of poultry intake reduced by 11% and 3% the risk of CRC incidence and mortality, respectively [68]. Each 120 g daily increment intake of poultry reduced the risk of breast cancer incidence by 3% [41]. The heterogeneity between the studies was below 50%.

Nonlinear dose-response analyses. No evidence of a nonlinear association between poultry intake and CRC incidence was found: along with the increment in poultry intake up to ~90 g/day the RRs decreased up to ~15% [68]. No association was found between poultry intake and risk of breast cancer incidence [41].

#### 2.2.17. Wine

A dose-response meta-analysis of seven prospective studies evaluating the associations between alcohol intake and CVD mortality (fatal CVD, fatal CHD, fatal ischemic heart disease) showed a J-shaped curve with a maximal reduction in RRs of 34% at 24 g/day of alcohol intake from wine [69]. However, the reduction in risk of ~34% was similar for intakes from ~10 to 30 g of alcohol from wine.

A dose-response meta-analysis between wine intake and T2DM risk, carried out on 13 prospective studies, showed a U-shaped association: all levels of daily wine intakes < 80 g were associated with a risk reduction and the lowest risk (20% of reduction) was evident at 20–30 g of daily intake of wine [70].

## 3. Discussion

### 3.1. Whole Grains

The intake of whole grains had a protective effect toward CVD [16], CHD [16,17], stroke [16], T2DM [18], and CRC risk [19,21]. Overall, a whole grain intake above 210 g/day is not necessary to obtain benefits on CVD, CHD, stroke, T2DM, and CRC risk.

Biological plausibility of such protective actions should be found in a beneficial effect of whole grain toward cardio-metabolic risk factors. RCTs and its meta-analyses showed a beneficial effect of whole grains compared with refined grains against CVD risk factors such as systolic and pulse blood pressure in healthy persons [71], diastolic blood pressure in overweight and obese adults [72], total and LDL cholesterol in healthy individuals [73], post-prandial blood glucose and insulin and the maximal glucose and insulin response in healthy subjects [74], post-prandial blood glucose and peripheral insulin resistance in obese adults [75], low-grade inflammation in overweight and obese subjects [76]. Fiber and many bioactive components in bran and in the germ are involved in this protective activity [77].

The MDPPI suggests 1 or 2 servings, three times a day. We propose that each serving of whole grain is 30 g with a total amount of 90–180 g/day. Wholemeal wheat sourdough bread, stoneground heat bread, wholemeal pasta, brown rice, whole grain cereals should be eaten every day. The intake of whole grains should be made by substituting food based on refined flours in order to avoid the increase of daily energy intake [78]. The intake of whole grain sourdough bread typical of the Mediterranean Diet of the early 1960s in Nicotera and maybe in Crete and Corfu is particularly recommended for its low glycemic index (GI) (revised in [9]). Italian law allows defining, as whole grain food, the products obtained by whole grain flour as well as the ones derived from refined flour added with bran or middling but it establishes that in this last case the single components (flour, bran and middling) should be clearly indicated [79]. Therefore, consumers’ choice should be made through a careful exam of the labels on foodstuffs. In the case of whole grain bread bought in bakeries, Italian law does not enforce the declaration of the quantities of the ingredients so it is impossible to establish the quantity of whole grain flour in the final product or whether it is a compound product of refined flour added with bran or middling [80].

### 3.2. Vegetables and Fruits

#### 3.2.1. Vegetables

In the nonlinear dose-response analyses, the highest reduction in RRs for CVD was reported at intakes of 600 g/day [24,29]. The maximal reductions in RRs of CHD were evident at 550–600 g/day [17,24] and of stroke at 500 g/day [24], respectively. The benefits on T2DM risk were minimal and obtained at intakes up to ~200–300 g/day [18,27]. The benefits on CRC risk were found up to ~200 g/day of intakes and not above [21].

The MDPPI suggests at least 2 servings of vegetables during the three main meals or, as an alternative, in the breaks between the main meals as snack. We propose that each serving of vegetables in the MDPPI is 100 g.

#### 3.2.2. Fruits

In dose-response analysis the associations between daily fruit intakes and CVD [24,29], CHD [17,24,25], stroke [17,24,30], T2DM [18,26,27,31], and CRC [21] had the highest reductions in RRs at intakes of ~200–300 g/day but for CVD risk further benefits were evident up to ~500–800 g/day of intakes [24,29].

The MDPPI recommends 1–2 servings, three times a day. We propose that each serving of fruits is 100 g.

The protective mechanisms of increasing vegetable and fruit intake toward CVD risk include decreasing blood pressure, regulation of lipids metabolism, reducing oxidative stress and low-grade inflammation [81,82,83]. The high content of antioxidants (flavonoids, vitamin C, Vitamin E, ß-carotene) reduces DNA damaging [84]. The protective effects of fruit intake toward T2DM depend on their richness in fiber that improves insulin sensitivity and reduces the risk of weight gain [31].

An increased intake of fruits and vegetables should be encouraged in their quantities as well as in their variety since different colors ensure the provision of different micronutrients in a well balanced diet [85]. The protective effect of the intake of fruits and vegetables over the risk of CVD depends on the richness in fiber, vitamins, minerals, phytochemicals which ensures antioxidant and anti-inflammatory effects as well as low glycemic load (GL) and energetic density [85].

### 3.3. Dairy (Milk, Cheese, Yogurt)

#### 3.3.1. Milk

According to a nonlinear dose-response meta-analysis, a daily intake of ~100 mL of milk reduces the risk of stroke by ~12% [42]. The protective effect of skim milk intake toward T2DM or breast cancer incidence was more evident at greater intakes of ~600 g/day [36,41]. Also, the protective effect of milk intake toward CRC was more evident at greater intakes of ~500–800 g [39].

#### 3.3.2. Cheese

In dose-response analysis the protective effect of cheese intake toward CVD, stroke and T2DM was evident for little intakes up to ~40 g/day [44], ~25 g/day [35], ~50 g/day [36], respectively without benefits for greater intakes.

#### 3.3.3. Yogurt

The protective effect of little daily quantities of yogurt intake was evident toward T2DM. The highest reductions in RRs were found up to 120–140 g/day [36] or 80 g/day [38] without benefits for higher intakes.

Overall these data indicated that small daily quantities of dairy products could have a protective effect toward CVD, stroke, and T2DM risk. The protective effect of milk and low-fat dairy on stroke risk could be in relation to a reduced incidence of hypertension [86]. It is worth noting that in a number of cheeses many angiotensin I-converting enzyme inhibitory peptides were identified [87,88].

The MDPPI suggests 2–3 servings, a day. We propose that 1 serving of milk is 50 g, 1 serving of yogurt is 50 g, 1 serving of cheese is 30 g.

Low-fat dairy should be preferably chosen. There is evidence that low-fat dairy has a better beneficial effect than high-fat dairy in the prevention of T2DM [89] and of CVD mortality [90]. A review of a number of RCTs concluded that there was not enough evidence of an unfavorable effect of dairy products on the cardio-metabolic risk factors (lipids, blood pressure, inflammation, insulin resistance, vascular function) independently from their fat content, and that the possible unfavorable effect of saturated fatty acids could be nullified if they are consumed in the frame of the dairy food matrix [91].

Moreover, it should be considered that, in the evaluation of the cardio-metabolic effects of dairy food in the RCTs, the quality of diet of the control group is of fundamental importance. For instance, a recent RCT evaluated the cardio-metabolic effects of 5 iso-energetic diets (cheese, butter, carbohydrate, monounsaturated fatty acids and polyunsaturated fatty acids diets), and found that LDL cholesterol was significantly lower after the cheese diet compared to the butter diet, but significantly higher than the carbohydrate, the monounsaturated fatty acids and the polyunsaturated fatty acids diets. No meaningful difference was noted among the 5 diets on risk factors such as inflammation markers, blood pressure and glucose-insulin homeostasis [92]. The matrix effect in this study is limited to the metabolism of cholesterol for the lack of increase of LDL cholesterol after cheese intake compared to butter. Indeed, the calcium contained in cheese can link the saturated fatty acids in the intestine and increase their fecal excretion [92].

### 3.4. Nuts

In dose-response analyses the maximal reduction in risk were observed at ~15 g/day for CVD [46], at ~15–20 g/day [46] or at ~10–15 g/day [17] for CHD. Similarly, the maximal risk reduction was observed at ~10–15 g/day [46] or at 12 g/day [48] for stroke risk.

A favorable effect of nuts on CVD health is biologically plausible considering the unique composition of these monounsaturated fatty acids and polyunsaturated fatty acids, fiber, magnesium, arginine and polyphenols rich food [93]. Possible effects include a reduction of low-grade inflammation, oxidative stress, endothelial dysfunction and an improvement of the lipid profile and of insulin resistance [94].

The MDPPI recommends 1 or 2 servings, of nuts a day. We propose that 1 serving of nut is 15 g.

### 3.5. Olive Oil

The dose-response analysis showed a protective effect of olive oil intake toward T2DM risk. It decreased to 13% with increasing intake up to ~15–20 g/day and no benefits were evident above these intakes [18].

Olive oil has been defined as the hallmark of Mediterranean Diet [49]. In Nicotera, in the late 1950s, the intake of olive oil provided 13–17% of total daily energy [95].

A meta-analysis of RCTs supported a cardiovascular protective role of olive oil for its beneficial effects on low-grade inflammation and endothelial function [96]. The biological plausibility of a protective effect on T2DM risk depends on some extra virgin olive oil components. Indeed, the monounsaturated fatty acids (compared to the saturated fatty acids) [97] and the polyphenols improve insulin sensitivity in many ways and therefore the T2DM risk [98].

The MDPPI suggests an intake of 3–4 servings of extra virgin olive oil, per day. We propose that 1 serving of extra virgin olive oil is 10 g.

### 3.6. Spices and Herbs

No dose-response meta-analyses were identified for spice and herb intake and CVD, CHD, stroke, T2DM, CRC and breast cancer.

Because of their high phenolic component content, spices and herbs have antioxidant power, anti-inflammatory and anti-mutagen properties, and therefore they can have a role in the prevention of CVD, T2DM, cancer and other degenerative diseases that have the oxidative stress as an important cause [99]. Not only does their use add flavor to food but also contributes to the decrease of salt intake. The WHO recommends a reduction of < 5 g/day salt (<2 g/day sodium) in order to reduce blood pressure and CVD, CHD and stroke risks [100].

The MDPPI recommends a daily intake of herbs and spices.

### 3.7. Fish and Shellfish

The dose-response association between fish intake and CHD mortality had the maximal reduction in RRs at ~30–60 g of daily fish intake without any further benefits for lower and higher intakes [52]. In another meta-analysis, a daily intake of fish as little as 20 g reduced the CHD mortality of 7% [54].

In a broad review that evaluated the relationship between fish or fish oil intake and CHD mortality in prospective cohort studies and RCTs, the intake of 250 mg/day of eicosapentaenoic acid and docosahexaenoic acid reduced CHD mortality by 36% without further reductions for higher intakes [101]. These intakes are easily obtained with two-100 g servings a week of which at least one is blue fish (www.ieo.it/bda).

The n-3 polyunsaturated fatty acids of fish reduce the CVD risk with an anti-inflammatory, antiarrhythmic and antiplatelet aggregation effect [102]. An RCT meta-analysis reported that the intake of oily fish was associated with a significant reduction of plasma triglycerides and to a significant increase of HDL cholesterol [103]. Moreover, an RCT meta-analysis did not highlight any n-3 polyunsaturated fatty acids effect on insulin sensitivity [104].

The MDPPI suggests a fish intake ≥2 servings a week (preferably fatty fish). We propose that 1 serving of fish or shellfish is 100 g.

### 3.8. Legumes

A protective effect of legume intake on CHD risk was evident up to ~100 g/day without benefits for higher intakes [17].

Legumes include lentils, beans, chickpeas, peas, peanuts, soya and other podded plants [105]. Beans, lentils, chickpeas and peas are high in fiber, protein and low in fat and are also called pulses [106]. The above studies do not allow a differentiation between fresh and dry legumes.

The MDPPI recommends an intake of ≥2 servings a week of legumes. We propose that 1 serving of fresh legumes is 100 g and 1 serving of pulses is 50 g.

Coherently to the Mediterranean tradition, legumes should partially replace protein food of animal origin.

### 3.9. Eggs

No association between egg intake and CHD or stroke risk was found (up to 1 egg/day) whereas the association between egg intake and breast cancer showed an increment of the risk above an intake of 5 eggs/week (Table 11). The biological plausibility of a lack of an unfavorable effect of a higher intake of eggs over the risk of CHD and stroke is provided by RCTs that did not find a worsening in CVD risk markers (lipid profile, body weight) in greater consumers of eggs (up to 1 egg/day) [107]. An increase of the intake of cholesterol with the diet does not have a negative impact on the lipid profile because in about 75% of the population leads to a reduction of the absorption of the same and/or of its synthesis and therefore a moderate or absent difference in serum cholesterol (normal or hypo-responders subjects) [108]. In hyper-responding subjects, the dietary increase of cholesterol leads to an increase of LDL cholesterol but also of HDL cholesterol with minimal effects on the LDL/HDL ratio [109]. The saturated fatty acids and the trans-fatty acids represent the major determiners of the total and LDL cholesterol [110]. The increased risk of breast cancer at egg intake >5/week could happen in subjects whose serum cholesterol levels are influenced by dietary intake and an excess of cholesterol may increase the risk of breast cancer through an increase of sexual hormones that promote cellular proliferation [57].

The MDPPI advised an intake of 2–4 eggs/week.

### 3.10. Refined Grains

The nonlinear dose-response analyses indicated that the RRs of CHD [17] and T2DM [18] were greater than 1.00 for intakes higher than ~100–120 g/day. The risk of T2DM increased by 6–14% with an intake of 200–400 g/day of refined grains [18].

The biological plausibility of an absence of a protective effect of refined grains respect to the whole ones against CHD, stroke and T2DM depends on the removal of fiber, micronutrients and minerals following the elimination of the bran and germ with cardio-metabolic protective effects [111]. Just like white potatoes and added sugar, refined cereals have a high GI and can increase the GL of the diet. They produce rapid glycemic and insulinemic peaks after the intake making way to adverse events such as stimulating reward/craving in cerebral areas, activation of the hepatic de novo lipogenesis and enabling visceral adiposity (reviewed in [111]). A recent international consensus established that low GI/GL diets reduce the risk of T2DM development in both sexes and they also reduce CHD risk mostly in women. These protective effects have greater relevance in sedentary, overweight or insulin resistant subjects. Low GI diets could have a protective effect against some cancer types such as CRC and breast cancer (reviewed in [112]).

The MDPPI recommends the limit of intake of refined grains of 3 servings a week. We propose that 1 serving of refined cereals is 60 g.

### 3.11. Potatoes

In nonlinear dose-response meta-analyses no association was found between daily potato intake and CHD or stroke, instead, the association between potato intake and T2DM or CRC showed increased risk above ~134 g/day. The increased risk of T2DM was mainly due to French fries intakes [58].

The MDPPI recommends the limit of intake of potatoes of 3 servings a week. We propose that one serving of potatoes is 100 g.

### 3.12. Red Meat and Processed Meat

#### 3.12.1. Red Meat

In nonlinear dose-response meta-analyses, a red meat intake up to 100 g/day increased the CHD risk by 20% and of stroke by 10% [17]. The association between red meat intake and increased risk of T2DM [18], CRC [21] and breast cancer [41] was linear.

The MDPPI recommends the intake of ≤2 servings of red meat a week. We propose that 1 serving of red meat is 100 g.

#### 3.12.2. Processed Meat

In nonlinear dose-response meta-analyses, the association between processed meat intake and CHD was null [17]. Otherwise the risk of stroke [17], T2DM [18], CRC [21] increased by 15%, 30%, and 20% with increased intakes above 70 g/day, 50 g/day and 60 g/day, respectively.

The MDPPI recommends an intake of processed meat of ≤1 serving per week. We propose that 1 serving is 50 g.

The mechanisms that link the intake of meat to NCDs risk involve a number of substances.

Haem iron has a pro-oxidant action that increases oxidative stress. The saturated fatty acids (processed meat is particularly rich) increase LDL cholesterol. Advanced glycation end products (which are found in animal products rich in fats and proteins especially if processed) have a pro-inflammatory action. Nitrates and nitrites (used in meat preservation) facilitate endothelial dysfunction, atherosclerosis and insulin resistance. Salt (used in meat preservation) has a hypertensive effect and it is a possible risk factor for gastric cancer. Polycyclic aromatic hydrocarbons and heterocyclic aromatic amines (which are formed during meat cooking at high temperatures) are carcinogenic especially for CRC (reviewed in [113,114,115]). A greater intake of red meat was in relation to a higher risk of CRC cancer. In 2015 the International Agency for Research on Cancer classified the intake of red meat as “probably carcinogenic for humans” and processed meat as “carcinogenic for humans” [116].

### 3.13. Poultry

In nonlinear dose-response meta-analyses, an intake of ~90 g/day of poultry reduced CRC incidence by ~15% [68].

The MDPPI suggests that poultry intake should be ≤2 servings a week. We propose that one serving is 100 g.

### 3.14. Sweets and Cakes and Cookies

No dose-response meta-analyses were identified for sweet/cake/cookie intakes.

The vast majority of sweets contain carbohydrates, which are rapidly digested such as refined flours and high GI sugar, which can increase the GL diet. They are often rich in industrial trans fatty acids [117] that increase CHD and sudden death risk because of the unfavorable effects on the lipid profile, insulin resistance, visceral adiposity, endothelial inflammation and dysfunction [111]. An occasional intake of sweets probably does not have a negative impact on health as opposed to a regular intake.

The MDPPI recommends an intake of ≤2 servings of sweets a week. We propose that 1 serving is 25 g. This quantity was arbitrarily decided.

### 3.15. Wine

Based on 2 dose-response meta-analyses the greater protection against CVD mortality was evident at 15–30 g of daily intake of alcohol from wine [69], and against T2DM risk at 20–30 g/day of wine [70].

The cardio-metabolic protective effects of light to moderate alcohol intake depend on an increase of HDL cholesterol and of adiponectin, a reduction of the low-grade inflammation, an improvement of insulin sensitivity, and endothelial function (reviewed in [118]).

The MDPPI suggests a moderate intake of red wine during meals. We propose that 1 serving is 10 g of ethanol and up to 3 serving and up to 1 serving and half were indicated for men and women, respectively.

Although these quantities are obtained from a little number of meta-analyses, several reviews indicated similar intakes of alcohol as optimal [118,119,120]. A regular and moderate wine intake during meals is a “Mediterranean way of drinking” [121].

## 4. Second Section

### Methods and Results

Based on frequencies and serving sizes indicated in Table 17 we built a weekly menu plan (Appendix A).

For some food groups, we calculated equivalents portions: 30 g-whole grain servings were 30 g of wholemeal pasta or whole rice or whole barley or whole spelled, 40 g of wholemeal sourdough bread, 20 g of breakfast cereals; 60 g-refined grain servings were 60 g of white pasta, 80 g of white bread; 100 g-fruit servings were 70 g of figs or grapes or prickly pears and 130 g of apricots or peaches or medlars. Nutritional diet composition is in Table 18.

## 5. Discussion

The diet built according the advice of MDPPI (Table 17) was very similar to that of Nicotera in the late 1950s that has been chosen as Italian Reference Mediterranean Diet [95] with the exception of percentage of energy provided by cereals that was lower and of fruits and vegetables that was higher. Indeed, Fidanza reported the following percentages of total energy of food groups: cereal (50–59%), virgin olive oil (13–17%), vegetables (2.2–3.6%), potatoes (2.3–4.4%), legumes(3–6%), fruits (2.6–3.6% including nuts that were about 3% of the weight of all fruits), fish (1.6–2.0%), red wine (1–6%), meat (2.6–5.0%), dairy (2–4%). The intake of eggs and animal fats was rare [95]. Saturated fatty acids were only the 6% of daily energy intake. Also the MAI was very close to the MAI of the diet in 1960 that was 9.4 for males and 11.4 for females living in Nicotera [122].

## 6. Conclusions

The MDPPI represents a modification proposal of the Modern Mediterranean Diet Pyramid presented during the third conference of CIISCAM (Centro Interuniversitario Internazionale di Studi sulle Culture Alimentari Mediterranee) in Parma, Italy, on November 3, 2009 (www.inran.it). In the MDPPI the modification interested cereal food, which were divided into two groups. The whole grain cereal derived foods, which are at the base of the pyramid and the foods derived from refined flour, which are at the top of the pyramid along with other food to be consumed with moderation. The presence at the bottom of the pyramid of wholemeal wheat sourdough bread and stoneground wheat bread is remarkable because they were typically used in Nicotera in the beginning of the 1960s and probably also in Crete and Corfu. They are rich in fiber and have a low GI (reviewed in [9]). Refined cereal foods are placed at the top of the pyramid for their possible unfavorable cardio-metabolic effects [111]. According to a recent survey the consumption of whole grain foods in Italy is quite low [123].

The present study is a systematic review of dose-response meta-analyses of prospective studies, which evaluated the relationship among food groups and selected nutrition-related NCDs such as CVD including CHD and stroke, T2DM, CRC and breast cancer. Our purpose has been to derive from these meta-analyses serving sizes of food group belonging to MDPPI with a protective (Mediterranean food) or a non-adverse (non-Mediterranean food) effect toward the above mentioned NCDs. Subsequently we have evaluated the compatibility of a weekly menu plan built according the MDPPI advice, with the Nicotera diet in the late 1950s. Only for sweets/cakes/cookies the lack of dose-response meta-analyses led us to define an arbitrarily serving size. In our opinion, the advice of the MDPPI as dietary pattern well defined into its characteristics as quantity and frequency of intake is compatible with the Reference Italian Mediterranean Diet. It is a plant-based dietary pattern [124] rich in high quality plant food that lowers the risk of T2DM [125] and CHD [126].

However, our study has some limitations. A first limit is given by the fact that the distinction between some food groups is not precise and there are overlaps. For example, the vegetable group includes beans, peas, potatoes, herbs as onions and garlic and so on; the legumes group includes peanuts that are often included in the nuts group. Another limit is that the single items of food groups could be not identical to those typical of Mediterranean area. The level of evidence that is variable in several meta-analyses depending on the quality of the studies considered, the strength of the associations and the presence of heterogeneity represent another limit.

All in all, we think that the MDPPI can represent a valid instrument for the definition of Mediterranean Diet that must also consider the types of food, the amounts, and their intake frequency. The indicated amounts are compatible both with the studies which evaluated the effect of the intake of food groups on different diet-correlated NCDs outcomes and the ones of the Italian Reference Mediterranean Diet of the late 1950s.

## Figures and Tables

**Figure 1 nutrients-11-01296-f001:**
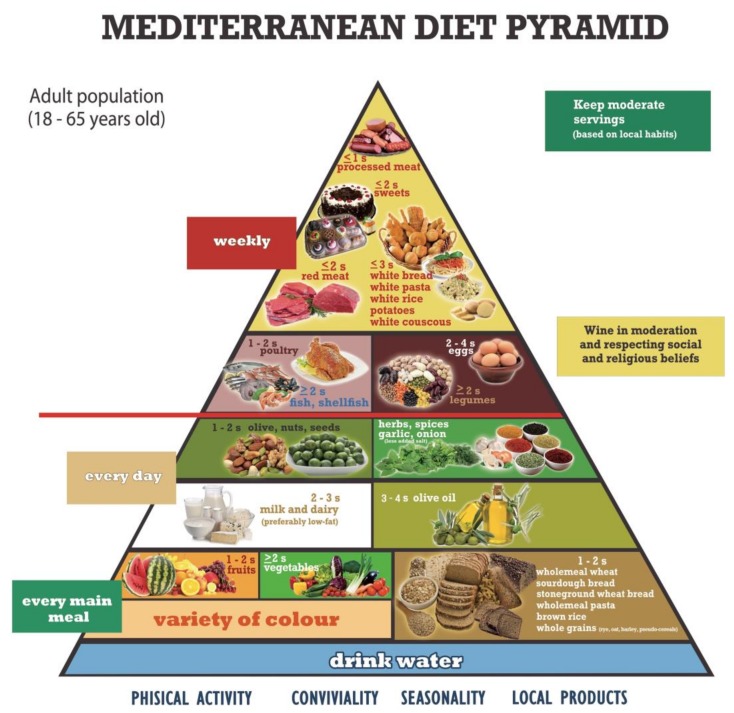
Proposal of Mediterranean Diet Pyramid for Italian People [9].

**Table 1 nutrients-11-01296-t001:** Summary of linear dose-response meta-analyses of prospective studies on whole grain intake and CVD, CHD, stroke, T2DM and CRC.

Authors, Year, Reference	No. of Studies	Each Increment Intake Per Day	RR (95% CI)	I^2^ Statistic	*p*-Value for Heterogeneity	Begg’s or Egger’s Test *p*-Value
Benisi-Kohansal 2016 [12]	3	90 g	CVD mortality 0.83 (0.76–0.91) *	0.00%	0.860	NR
Chen 2016 [13]	10	50 g	CVD mortality 0.70 (0.61–0.79)	64.80%	0.002	0.370/0.140
Li 2016 [14]	8	30 g	CVD mortality 0.95 (0.92–0.98)	68.60%	<0.001	0.276/0.202
Wei 2016 [15]	8	90 g	CVD mortality 0.74 (0.66–0.83)	76.30%	<0.001	0.107/0.834
Aune 2016 [16]	10	90 g	CVD 0.78 (0.73–0.85)	40.00%	0.090	0.310
Bechthold 2017 [17]	5	30 g	CHD 0.95 (0.92–0.98)	46.00%	0.110	NR
Aune 2016 [16]	7	90 g	CHD 0.81 (0.75–0.87)	9.00%	0.360	0.110
Bechthold 2017 [17]	4	30 g	Stroke 0.99 (0.95–1.03)	65.00%	0.040	NR
Aune 2016 [16]	6	90 g	Stroke 0.88 (0.75–1.03)	56.00%	0.040	0.010
Schwingshackl 2017 [18]	12	30 g	T2DM 0.87 (0.82–0.93)	91.00%	NR	NR
Aune 2011 [19]	6	90 g	CRC 0.83 (0.78–0.89)	18.00%	0.300	1.000/0.540
Vieira 2017 [20]	6	90 g	CRC 0.83 (0.79–0.89)	18.00%	0.300	NS
Schwingshackl 2018 [21]	9	30 g	CRC 0.95 (0.93–0.97)	58.00%	0.0200	NR

* specific whole grains; CVD, cardiovascular disease; CHD, coronary heart disease; T2DM, type 2 diabetes mellitus; CRC, colorectal cancer; RR, relative risk; NR, not reported; NS, not significant.

**Table 2 nutrients-11-01296-t002:** Summary of linear dose-response meta-analyses of prospective studies on vegetable intake and CVD, CHD, stroke, T2DM, CRC, breast cancer.

Authors, Year, Reference	No. of Studies	Each Increment Intake Per Day	RR (95% CI)	I^2^ Statistic	*p*-Value for Heterogeneity	Begg’s or Egger’s Test *p*-Value
Aune 2017 [24]	14	200 g	CVD 0.90 (0.87–0.93)	11.50%	0.330	0.530
Gan 2015 [25]	13	400 g	CHD 0.82 (0.73–0.92)	35.60%	0.068	0.880/0.381
Aune 2017 [24]	20	200 g	CHD 0.84 (0.79–0.90)	60.60%	<0.0001	0.001
Bechthold 2017 [17]	14	100 g	CHD 0.97 (0.96–0.99)	12.00%	0.320	NR
Aune 2017 [24]	13	200 g	Stroke 0.87 (0.79–0.96)	63.40%	0.001	0.150
Bechthold 2017 [17]	10	100 g	Stroke 0.92 (0.86–0.98)	79.00%	<0.001	NR
Li 2014 [26]	5	106 g	T2DM 0.98 (0.89–1.08)	45.80%	0.117	0.117
Wu 2015 [27]	7	106 g	T2DM 0.98 (0.95–1.01)	78.30%	0.000	0.130/0.150
Schwingshackl 2017 [18]	11	100 g	T2DM 0.98 (0.96–1.00)	62.00%	NR	NS
Vieira 2017 [20]	11	100 g	CRC 0.98 (0.96–0.99)	0.00%	0.480	NS
Schwingshackl 2018 [21]	15	100 g	CRC 0.97 (0.96–0.98)	0.00%	0.640	0.530
Aune 2012 [28]	9	200 g	Breast cancer 1.00 (0.95–1.06)	17.00%	0.290	NR

CVD, cardiovascular disease; CHD, coronary heart disease; T2DM, type 2 diabetes mellitus; CRC, colorectal cancer; RR, relative risk; NR, not reported; NS, not significant.

**Table 3 nutrients-11-01296-t003:** Summary of linear dose-response meta-analyses of prospective studies on fruit intake and CVD, CHD, stroke, T2DM, CRC, breast cancer.

Authors, Year, Reference	No. of Studies	Each Increment Intake Per Day	RR (95% CI)	I^2^ Statistic	*p*-Value for Heterogeneity	Begg’s or Egger’s Test *p*-Value
Aune 2017 [24]	17	200 g	CVD 0.87 (0.82–0.92)	79.10%	<0.0001	0.410
Gan 2015 [25]	15	300 g	CHD 0.84 (0.75–0.93)	31.70%	0.0780	0.367/0.591
Aune 2017 [24]	24	200 g	CHD 0.90 (0.86–0.94)	43.70%	0.0100	0.040
Bechthold 2017 [17]	13	100 g	CHD 0.94 (0.90–0.97)	71.00%	<0.0010	NR
Aune 2017 [24]	16	200 g	Stroke 0.82 (0.74–0.90)	72.90%	<0.0001	0.620
Bechthold 2017 [17]	10	100 g	Stroke 0.90 (0.84–0.97)	86.00%	0.0010	NR
Li 2014 [26]	7	106 g	T2DM 0. 94 (0.89–1.00)	0.00%	0.059	NR
Wu 2015 [27]	9	106 g	T2DM 0.99 (0.97–1.00)	18.60%	0.278	0.470/0.680
Schwingshackl 2017 [18]	13	100 g	T2DM 0.98 (0.97–1.00)	21.00%	NR	NS
Vieira 2017 [20]	13	100 g	CRC 0.96 (0.93–1.00)	68.00%	<0.0001	NS
Schwingshackl et al. 2018 [21]	16	100 g	CRC 0.97 (0.95–0.99)	61.00%	<0.0010	0.120
Aune 2012 [28]	10	200 g	Breast cancer 0.94 (0.89–1.00)	39.00%	0.1000	NR

CVD, cardiovascular disease; CHD, coronary heart disease; T2DM, type 2 diabetes mellitus; CRC, colorectal cancer; RR, relative risk; NR, not reported; NS, not significant.

**Table 4 nutrients-11-01296-t004:** Summary of linear dose-response meta-analyses of prospective studies on milk intake and CVD, CHD, stroke, T2DM, CRC, and breast cancer.

Authors, Year, Reference	No. of Studies	Each Increment Intake Per Day	RR (95% CI)	I^2^ Statistic	*p*-Value for Heterogeneity	Begg’s or Egger’s Test *p*-Value
Soedamah-Muthu 2011 [32]	4	200 mL	CVD 0.94 (0.89–0.99)	0.00%	0.5020	NR
Guo 2017 [33]	12	244 g	CVD 1.01 (0.93–1.10)	92.40%	<0.0010	0.449
Soedamah-Muthu 2011 [32]	6	200 mL	CHD 1.00 (0.96–1.04)	26.90%	0.2330	NR
Mullie 2016 [34]	9	200 mL	CHD 1.01 (0.98–1.05)	16.00%	0.3000	0.680/0.050
Guo 2017 [33]	12	244 g	CHD 1.01 (0.96–1.06)	45.50%	0.0430	0.397
Soedamah-Muthu 2011 [32]	6	200 mL	Stroke 0.87 (0.72–1.07)	94.60%	0.0000	NR
de Goede 2016 [35]	14	200 g	Stroke 0.93 (0.88–0.98)	86.00%	<0.0010	0.060
Mullie 2016 [34]	10	200 mL	Stroke 0.91 (0.82–1.02)	92.00%	<0.0100	0.530/0.050
Aune 2013 [36]	7	200 g	T2DM 0.87 (0.72–1.04)	93.60%	<0.0001	0.410
Gao 2013 [37]	8	200 g	T2DM incidence 0.89 (0.79–1.01)	66.30%	0.0050	NR
Gijsbers 2016 [38]	12	200 g	T2DM incidence 0.97 (0.93–1.02)	57.40%	0.0070	0.071
Aune 2012 [39]	9	200 g	CRC 0.90 (0.85–0.94)	0.00%	0.6200	0.840/0.860
Vieira 2017 [20]	9	200 g	CRC 0.94 (0.92–0.96)	0.00%	0.9700	NS
Dong 2011 [40]	9	200 g	Breast cancer 0.98 (0.95–1.01)	NR	>0.3000	>0.050
Wu 2016 [41]	11	200 g	Breast cancer incidence 0.97 (0.93–1.01)	36.40%	NR	0.436/0.355

CVD, cardiovascular disease; CHD, coronary heart disease; T2DM, type 2 diabetes mellitus; CRC, colorectal cancer; RR, relative risk; NR, not reported; NS, not significant.

**Table 5 nutrients-11-01296-t005:** Summary of linear dose-response meta-analyses of prospective studies on cheese intake and CVD, CHD, stroke, T2DM, and CRC.

Authors, Year, Reference	No. of Studies	Each Increment Intake Per Day	RR (95% CI)	I^2^ Statistic	*p*-Value for Heterogeneity	Begg’s or Egger’s Test *p*-Value
Chen 2017 [44]	7	50 g	CVD 0.92 (0.83–1.02)	16.90%	0.301	>0.100
Guo 2017 [33]	11	10 g	CVD 0.98 (0.95–1.00)	82.60%	<0.001	NR
Chen 2017 [44]	8	50 g	CHD 0.90 (0.84–0.95)	0.00%	0.444	0.170/0.040
Guo 2017 [33]	10	10 g	CHD 0.99 (0.97–1.02)	40.30%	0.089	0.273
Chen 2017 [44]	5	50 g	Stroke 0.94 (0.84–1.04)	63.70%	0.026	>0.10
de Goede 2016 [35]	7	40 g	Stroke 0.97 (0.94–1.01)	31.20%	0.179	NR
Aune 2013 [36]	8	50 g	T2DM 0.92 (0.86–0.99)	0.00%	0.790	0.740
Gao 2013 [37]	7	30 g	T2DM incidence 0.80 (0.69–0.93)	59.00%	0.020	NR
Gijsbers 2016 [38]	13	10 g	T2DM incidence 1.00 (0.99–1.02)	61.70%	0.002	0.880
Aune 2012 [39]	7	50 g	CRC 0.96 (0.83–1.12)	28.00%	0.220	NR
Vieira 2017 [20]	7	50 g	CRC 0.94 (0.87–1.02)	10.00%	0.360	NS

CVD, cardiovascular disease; CHD, coronary heart disease; T2DM, type 2 diabetes mellitus; CRC, colorectal cancer; RR, relative risk; NR, not reported; NS, not significant.

**Table 6 nutrients-11-01296-t006:** Summary of linear dose-response meta-analyses of prospective studies on yogurt intake and CVD, CHD, stroke, T2DM, and breast cancer.

Authors, Year, Reference	No. of Studies	Each Increment Intake Per Day	RR (95% CI)	I^2^ Statistic	*p*-Value for Heterogeneity	Begg’s or Egger’s Test *p*-Value
Guo 2017 [33]	3	50 g	CVD 1.03 (0.97–1.09)	0.00%	0.499	NR
Guo 2017 [33]	3	50 g	CHD 1.03 (0.97–1.09)	0.00%	0.685	NR
de Goede 2016 [35]	3	100 g	Stroke 1.02 (0.90–1.17)	47.80%	0.147	NR
Aune 2013 [36]	7	200 g	T2DM 0.78 (0.60–1.02)	69.90%	0.003	0.370
Gao 2013 [37]	7	50 g	T2DM incidence 0.91 (0.82–1.00)	74.00%	0.001	NR
Gijsbers 2016 [38]	12	50 g	T2DM incidence 0.94 (0.90–0.97)	73.30%	0.000	0.180
Wu 2016 [41]	3	200 g	Breast cancer incidence 0.87 (0.72–1.06)	0.00%	NR	1.000/0.488

CVD, cardiovascular disease; CHD, coronary heart disease; T2DM, type 2 diabetes mellitus; RR, relative risk; NR, not reported.

**Table 7 nutrients-11-01296-t007:** Summary of linear dose-response meta-analyses of prospective studies on nut intake and CVD, CHD, stroke, T2DM, CRC, breast cancer.

Authors, Year, Reference	No. of Studies	Each Increment Intake Per Day	RR (95% CI)	I^2^ Statistic	*p*-Value for Heterogeneity	Begg’s or Egger’s Test *p*-Value
Luo 2014 [45]	4	28 g	CVD incidence 0.71 (0.59–0.85)	48.80%	0.119	0.090
Aune 2016 [46]	11	28 g	CVD 0.80 (0.72–0.89)	56.00%	0.001	NR
Grosso 2015 [47]	5	28 g	CVD mortality 0.61 (0.42–0.91)	75.00%	NR	NR
Aune 2016 [46]	11	28 g	CHD 0.71 (0.63–0.80)	47.00%	0.040	0.280
Bechthold 2017 [17]	4	28 g	CHD 0.67 (0.43–1.05)	85.00%	0.001	NR
Aune 2016 [46]	11	28 g	Stroke 0.93 (0.83–1.05)	14.00%	0.310	0.300
Bechthold 2017 [17]	6	28 g	Stroke 0.99 (0.84–1.17)	45.00%	0.110	NR
Luo 2014 [45]	4	28 g	T2DM incidence 1.03 (0.91–1.16)	63.90%	0.040	0.810
Schwingshackl 2017 [18]	7	28 g	T2DM 0.89 (0.71–1.12)	77.00%	NR	NR
Schwingshackl 2018 [21]	4	28 g	CRC 0.96 (0.76–1.21)	25.00%	0.260	NR
Wu 2016 [41]	3	28 g	Breast cancer incidence 0.96 (0.84–1.09)	0.00%	NR	0.100/0.955

CVD, cardiovascular disease; CHD, coronary heart disease; T2DM, type 2 diabetes mellitus; CRC, colorectal cancer; RR, relative risk; NR, not reported.

**Table 8 nutrients-11-01296-t008:** Summary of linear dose-response meta-analyses of prospective studies on olive oil intake and CHD, stroke, T2DM.

Authors, Year, Reference	No. of Studies	Each Increment Intake Per Day	RR (95% CI)	I^2^ Statistic	*p*-Value for Heterogeneity	Begg’s or Egger’s Test *p*-Value
Martínez-Gonzáles 2014 [49]	5	25 g	CHD 0.94 (0.78–1.14)	66.20%	0.020	NR
Martínez-Gonzáles 2014 [49]	3	25 g	Stroke 0.76 (0.67–0.86)	0.00%	0.440	0.110
Schwingshackl 2017 [50]	4	10 g	T2DM 0.91 (0.87–0.95)	0.00%	NR	NR

CHD, coronary heart disease; T2DM, type 2 diabetes mellitus; RR, relative risk; NR, not reported.

**Table 9 nutrients-11-01296-t009:** Summary of linear dose-response meta-analyses of prospective studies on fish intake and CVD, CHD, stroke, T2DM, CRC, breast cancer.

Authors, Year, Reference	No. of Studies	Each Increment Intake Per Day	RR (95% CI)	I^2^ Statistic	*p*-Value for Heterogeneity	Begg’s or Egger’s Test *p*-Value
Jayedi 2018 [51]	8	20 g	CVD mortality 0.96 (0.94–0.98)	0.00%	0.620	NR
Bechthold 2017 [17]	15	100 g	CHD 0.88 (0.79–0.99)	40.00%	0.060	NS
Zheng 2012 [52]	17	15 g	CHD mortality 0.94 (0.90–0.98)	63.10%	0.000	NR
Bechthold 2017 [17]	15	100 g	Stroke 0.86 (0.75–0.99)	25.00%	0.180	NR
Schwingshackl 2017 [18]	15	100 g	T2DM 1.09 (0.93–1.28)	84.00%	NR	NS
Vieira 2017 [20]	11	100 g	CRC 0.89 (0.80–0.99)	0.00%	0.520	NS
Schwingshackl 2018 [21]	16	100 g	CRC 0.93 (0.85–1.01)	12.00%	0.320	0.910
Zheng 2013 [53]	11	15 g	Breast cancer 1.00 (0.97–1.03)	64.00%	0.001	NS
Wu 2016 [41]	13	120 g	Breast cancer incidence 1.07 (0.94–1.21)	33.30%	NR	0.100/0.089

CVD, cardiovascular disease; CHD, coronary heart disease; T2DM, type 2 diabetes mellitus; CRC, colorectal cancer; RR, relative risk; NR, not reported; NS, not significant.

**Table 10 nutrients-11-01296-t010:** Summary of linear dose-response meta-analyses of prospective studies on legume intake and CHD, stroke, T2DM, CRC.

Authors, Year, Reference	No. of Studies	Each Increment Intake Per Day	RR (95% CI)	I^2^ Statistic	*p*-Value for Heterogeneity	Begg’s or Egger’s Test *p*-Value
Bechthold 2017 [17]	8	50 g	CHD 0.96 (0.92–1.01)	39.00%	0.120	NR
Bechthold 2017 [17]	6	50 g	Stroke 1.00 (0.88–1.13)	62.00%	0.020	NR
Schwingshackl 2017 [18]	12	50 g	T2DM 1.00 (0.92–1.09)	87.00%	NR	NR
Schwingshackl 2018 [21]	10	50 g	CRC 1.00 (0.92–1.08)	50.00%	0.040	0.590
Vieira 2017 [20]	4	50 g	CRC 1.00 (0.95–1.06)	33.00%	0.200	NS

CHD, coronary heart disease; T2DM, type 2 diabetes mellitus; CRC, colorectal cancer; RR, relative risk; NR, not reported; NS, not significant.

**Table 11 nutrients-11-01296-t011:** Summary of linear dose-response meta-analyses of prospective studies on egg intake and CHD, stroke, T2DM, breast cancer.

Authors, Year, Reference	No. of Studies	Each Increment Intake Per Day or Week	RR (95% CI)	I^2^ Statistic	*p*-Value for Heterogeneity	Begg’s or Egger’s Test *p*-Value
Rong 2013 [55]	9	1egg/day	CHD 0.99 (0.85–1.15)	0.00%	0.970	>0.050/>0.050
Rong 2013 [55]	8	1egg/day	Stroke 0.91 (0.81–1.02)	0.00%	0.460	>0.050/>0.050
Tamez 2016 [56]	13	1egg/day	T2DM 1.13 (1.04–1.22)	85.00%	<0.001	0.460
Keum 2015 [57]	6	5eggs/week	Breast cancer 1.05 (0.99–1.11)	0.00%	0.927	0.620

CHD, coronary heart disease; T2DM, type 2 diabetes mellitus; RR, relative risk.

**Table 12 nutrients-11-01296-t012:** Summary of linear dose-response meta-analyses of prospective studies on refined grain intake and CHD, stroke, T2DM.

Authors, Year, Reference	No. of Studies	Each Increment Intake Per Day	RR (95% CI)	I^2^ Statistic	*p*-Value for Heterogeneity	Begg’s or Egger’s Test *p*-Value
Bechthold 2017 [17]	4	30 g	CHD 1.01 (0.99–1.04)	0.00%	0.510	NR
Bechthold 2017 [17]	4	30 g	Stroke 1.00 (0.98–1.01)	0.00%	0.510	NR
Schwingshackl 2017 [18]	14	30 g	T2DM 1.01 (0.99–1.03)	59.00%	NR	NR

CHD, coronary heart disease; T2DM, type 2 diabetes mellitus; RR, relative risk; NR, not reported.

**Table 13 nutrients-11-01296-t013:** Summary of linear dose-response meta-analyses on potato intake and CHD, stroke, T2DM, CRC.

Authors, Year, Reference	No. of Studies	Total Intake Per Day	RR (95% CI)	I^2^ Statistic	*p*-Value for Heterogeneity	Begg’s or Egger’s Test *p*-Value
Schwingshackl 2018 [58]	7	150 g	CHD 1.03 (0.96–1.09)	0.00%	0.990	NR
Schwingshackl 2018 [58]	6	150 g	Stroke 0.98 (0.93–1.03)	3.00%	0.400	NR
Schwingshackl 2018 [58]	7	150 g	T2DM 1.18 (1.10–1.27)	30.00%	0.200	NR
Schwingshackl 2018 [58]	6	150 g	CRC 1.05 (0.92–1.20)	20.00%	0.280	NR

CHD, coronary heart disease; T2DM, type 2 diabetes mellitus; CRC, colorectal cancer; RR, relative risk; NR, not reported.

**Table 14 nutrients-11-01296-t014:** Summary of linear dose-response meta-analyses of prospective studies on red meat intake and CVD, CHD, stroke, T2DM, CRC, breast cancer.

Authors, Year, Reference	No. of Studies	Each Increment Intake Per Day	RR (95% CI)	I^2^ Statistic	*p*-Value for Heterogeneity	Begg’s or Egger’s Test, *p*-Value
Abete 2014 [59]	6	100 g	CVD mortality 1.15 (1.05–1.26)	76.60%	<0.001	>0.100/>0.100
Bechthold 2017 [17]	3	100 g	CHD 1.15 (1.08–1.23)	0.00%	0.680	NR
Bechthold 2017 [17]	7	100 g	Stroke 1.12 (1.06–1.17)	0.00%	0.500	NR
Aune 2009 [60]	9	120 g	T2DM 1.20 (1.04–1.38)	68.30%	0.001	NR
Feskens 2013 [61]	14	100 g	T2DM 1.13 (1.03–1.23)	36.00%	NR	NR
Schwingshackl 2017 [18]	14	100 g	T2DM 1.17 (1.08–1.26)	83.00%	NR	NS
Larsson 2006 [62]	14	120 g	CRC 1.28 (1.18–1.39)	0.00%	0.790	NR
Alexander 2011 [63]	13	70 g	CRC 1.05 (0.97–1.13)	NR	<0.001	0.97
Chan 2011 [64]	8	100 g	CRC incidence 1.17 (1.05–1.13)	0.00%	0.483	NS
Vieira 2017 [20]	8	100 g	CRC 1.12 (1.00–1.25)	24.00%	0.240	NS
Zhao 2017 [65]	9	100 g	CRC 1.16 (1.05–1.29)	0.00%	0.600	NR
Schwingshackl 2018 [21]	21	100 g	CRC 1.12 (1.06–1.19)	27.00%	0.130	0.620
Guo 2015 [66]	11	120 g	Breast cancer 1.11 (1.05–1.16)	NR	>0.100	NR
Wu 2016 [41]	8	120 g	Breast cancer incidence 1.13 (1.01–1.26)	56.40%	NR	0.266/0.110

CVD, cardiovascular disease; CHD, coronary heart disease; T2DM, type 2 diabetes mellitus; CRC, colorectal cancer; RR, relative risk; NR, not reported; NS, not significant.

**Table 15 nutrients-11-01296-t015:** Summary of linear dose-response meta-analyses of prospective studies on processed meat intake and CVD, CHD, stroke, T2DM, CRC, breast cancer.

Authors, Year, Reference	No. of Studies	Each Increment Intake Per Day	RR (95% CI)	I^2^ Statistic	*p*-Value for Heterogeneity	Begg’s or Egger’s Test *p*-Value
Abete 2014 [59]	6	50 g	CVD mortality 1.24 (1.09–1.40)	76.40%	0.0010	>0.100/>0.100
Wang 2016 [67]	10	50 g	CVD mortality 1.15 (1.07–1.24)	75.40%	<0.0100	≥0.370/≥0.540
Bechthold 2017 [17]	3	50 g	CHD 1.27 (1.09–1.49)	0.00%	0.5100	NR
Bechthold 2017 [17]	6	50 g	Stroke 1.17 (1.02–1.34)	56.00%	0.0500	NR
Aune 2009 [60]	8	50 g	T2DM 1.57 (1.28–1.93)	74.00%	<0.0001	NR
Feskens 2013 [61]	21	50 g	T2DM 1.32 (1.19–1.48)	89.00%	NR	NR
Schwingshackl 2017 [18]	14	50 g	T2DM 1.37 (1.22–1.55)	88.00%	NR	NR
Larsson 2006 [62]	11	30 g	CRC 1.09 (1.05–1.13)	0.00%	0.7800	NR
Chan 2011 [64]	9	50 g	CRC incidence 1.18 (1.10–1.28)	12.20%	0.3330	NS
Vieira 2017 [20]	10	50 g	CRC 1.18 (1.10–1.28)	11.00%	0.3400	NS
Zhao 2017 [65]	8	50 g	CRC 1.22 (1.12–1.33)	19.00%	0.2800	NR
Schwingshackl 2018 [21]	16	50 g	CRC 1.17 (1.10–1.23)	6.00%	0.3900	0.660
Guo 2015 [66]	7	50 g	Breast cancer 1.09 (1.03–1.16)	NR	>0.100	NR
Wu 2016 [41]	20	50 g	Breast cancer incidence 1.09 (1.02–1.17)	11.80%	0.3290	0.945/0.566

CVD, cardiovascular disease; CHD, coronary heart disease; T2DM, type 2 diabetes mellitus; CRC, colorectal cancer; RR, relative risk; NR, not reported; NS, not significant.

**Table 16 nutrients-11-01296-t016:** Summary of linear dose-response meta-analyses of prospective studies on poultry intake and CVD, CHD, stroke, T2DM, CRC, breast cancer.

Authors, Year, Reference	No. of Studies	Each Increment Intake Per Day	RR (95% CI)	I^2^ Statistic	*p*-Value for Heterogeneity	Begg’s or Egger’s Test *p*-Value
Feskens 2013 [61]	10	100 g	T2DM 1.04 (0.82–1.32)	51.00%	NR	NR
Shi 2015 [68]	16	50 g	CRC incidence 0.89 (0.81–0.97)	41.20%	0.043	0.140
Shi 2015 [68]	4	50 g	CRC mortality 0.97 (0.79–1.20)	0.00%	0.695	NR
Wu 2016 [41]	10	120 g	Breast cancer incidence 0.97 (0.85–1.11)	33.20%	NR	0.107/0.090

T2DM, type 2 diabetes mellitus; CRC, colorectal cancer; RR, relative risk; NR, not reported.

**Table 17 nutrients-11-01296-t017:** Mediterranean Diet Pyramid proposed for Italian people (MDPPI).

Food Groups	Frequency of Intake	Serving Size (g, mL, on Average)
Whole grains	1–2 servings every main meal (three meals)	30 g
Fruits	1–2 servings every main meal (three meals)	100 g
Vegetables	≥2 servings every main meal (three meals)	100 g
Milk and dairy	2–3 servings/day	Milk 50 mL
(preferably low-fat)		Yogurt 50 g
		Cheese 30 g
Nuts	1–2 servings/day	15 g
Extra Virgin Olive oil	3–4 servings/day	10 g
Herbs and spices	use them every day	
Fish and shellfish	≥2 servings/week	100 g
Poultry	1–2 servings/week	100 g
Legumes	≥2 servings/week	100 g fresh, 50 g dry
Eggs	2–4 servings/week	1 egg
Refined grains	≤3 servings/week	60 g
Potatoes	≤3 servings/week	100 g
Red meat	≤2 servings/week	100 g
Processed meat	≤1 serving/week	50 g
Sweets	≤2 servings/week	25 g
Red wine	≤2 glasses/day for men	15 g of alcohol
	≤1 glass/day for women	

**Table 18 nutrients-11-01296-t018:** Nutritional composition (mean of 7 days) and number of servings of weekly menu plan built on advice of Mediterranean Diet Pyramid for Italian People.

Energy intake: 1998.85 kcal/day	
Carbohydrates: 246.67 g (46.3%)	
Protein: 82.31 g (16.5%) vegetal protein 49.27 g and animal protein 30.64 g	
Fats: 71.08 g (32%): SFAs 13.26 g (6.1% of total kcal), MUFAs 36.13 g, PUFAs,11.82 g	
Fiber: 47.48 g	
Ethanol: 15 g	
Red Wine: 5.2% of total kcal	1 drink/day
Olive oil: 13.5% of total kcal	3 servings/day
Potatoes: 1.9% of total kcal	3 servings/week
Legumes: 6.7% of total kcal	1 serving/day
Refined grains: 3.9% of total kcal	3 servings/week
Whole grains: 27.4% of total kcal	2 servings at every main meal (6 servings/day)
Vegetables: 5.7% of total kcal	2 servings at every main meal (6 servings/day)
Fresh fruits: 11% of total kcal	2 servings at every main meal (6 servings/day)
Nuts: 9.1% of total kcal	2 servings/day
Processed meat: 0.9% of total kcal	1 serving/week
Red Meat: 1.4% of total kcal	2 servings/week
Poultry: 1.4% of total kcal	2 servings/week
Milk and dairy: 4.1% of total kcal	2 servings/day
Fish and shellfish: 3.5% of total kcal	3 servings/week
Eggs: 1.4% of total kcal	2 eggs/week
Sweets: 1.8% of total kcal	2 servings/week
MAI [77]: 8	
Average weekly GI: 46%	
Average weekly GL: 115.89	
Total calories provided from vegetable food: 77%	
Total calories provided from animal food: 23%	

SFAs, saturated fatty acids; MUFAs, monounsaturated fatty acids; PUFA, polyunsaturated fatty acids; MAI, Mediterranean Adequacy index; GI, glycemic index; GL, glycemic load.

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
