# Peer review of "Mediterranean Diet Pyramid: A Proposal for Italian People. A Systematic Review of Prospective Studies to Derive Serving Sizes"

_nutrients, 2019, doi:10.3390/nu11061296_

Reviewer 1 Report

Review of: Mediterranean Diet Pyramid: A Proposal for Italian People

The objective of the present literature study was 1) to systematically review dose-response meta-analyses of prospective studies with the aim to derive the quantities of food to consume to attain a protective effect or a non-unfavorable effect toward some noncommunicable diseases such as CVD, T2DM, colorectal (CRC) and breast cancer. 2) Furthermore a weekly menu plan was built on the advice about frequency of intakes and serving sizes and the bromatological composition of this diet was compared with the Reference Italian Mediterranean Diet followed in 1960 in Nicotera.

The topic is interesting and appealing to readers, as the adoption of MD is an effective public health strategy to tackle some noncommunicable diseases such as CVD, T2DM, colorectal (CRC) and breast cancer.

However, I advise the authors to improve the quality of presentation. Towards this direction the authors must revise the entire manuscript and the comments below should be taken into consideration when making the revision.

General comments

1.    Please consider presenting the various food groups in the same way they appear in figure 1.

2.    Please check headlines and subheadings. For example, in the "FIRST SECTION RESULTS" red and processed meat are presented in separate subsections (line 304 and line 322), while in "FIRST SECTION DISCUSSION" (line 558) they appear in the same subsection.

3.    If possible, combine the results and discussion section, in order to avoid repetition and redundancy. If this is not the case, try to be less wordy in the results section and highlight only the most important literature findings presented in the tables. For example, lines 307-321 can be characterized as a detailed description of all the information presented in Table 11A. Within this context, please take into consideration that although lines 560-563 can also be characterized as a repetition of the lines 307-321, they highlight the most important findings regarding red meat consumption.

4.    Please consider revising the "FIRST SECTION DISCUSSION" so as to be more accurate and concise and ensure that you avoid discussing the same biological mechanisms, i.e. in lines 394-398 and 413-417 since fruit and vegetables have a plethora of protective effects in common. Please take into consideration this comment when revising the whole manuscript.

Specific comments

Main manuscript

1.    Line 16: The word “effect” is repeated.

2.    Lines 17-19: Please rewrite for clarity.

3.    Lines 33-34: Please consider providing more details in order to be more informative.

4.    Lines 39-41: It would be quite interesting if further information was provided.

5.    Lines 43: Please change “Grece” to “Greece”.

6.    Line 45: “Those dietary habits” Taking into consideration ref [8], please rephrase in order to be more accurate.

7.    Lines 47-48: Please provide references in order to be more informative.

8.    Line 51: “to a variant of MDPPI”. Please rewrite for clarity.

9.    Lines 55-60: Please rewrite in order to be more accurate and concise.

10. Lines 129-134: Please rewrite for clarity.

11. Line 549: Please rephrase.

12. Line 556: Please rephrase.

13. Line 561: Based on data presented in lines 307-308, please check “20% and of stroke by 10%”.

14. Lines 597-598. As there was no dose-response meta-analyses identified for sweet/cake/cookie/ intakes, on what scientific bases did you proposed that 1 serving is equal to 25g? Please provide the appropriate references

15. Line 659: Please rephrase for clarity.

16. The lines mentioned below can be characterized as a detailed description of all the information presented in Tables. Please modify. In particular:

Lines 90-112

Table 1

Lines 116-143

Table 2A

Lines 147-171

Table 2B

Lines 175-190

Table 3A

Lines 194-205

Table 3B

Lines 209-215

Table 3C

Lines 219-236

Table 4

Lines 240-245

Table 5

Lines 249-262

Table 6

Lines 266-272

Table 7

Lines 276-281

Table 8

Lines 285-291

Table 9

Lines 295-303

Table 10

Lines 307-321

Table 11A

Lines 325-228

Table 11B

Lines 342-248

Table 11C

 Tables

1.    Please check data provided regarding ref [17]

2.    Table 12: Please consider moving table 12 to the supplementary materials.

3.    Table 13: Please explain in more details the methodology followed.

 Supplementary materials

1. Although “seasonality” and “local production” are emphasized in Figure 1, these parameters were not taken into consideration in the Weekly menu plan. Please comment on.

Author Response

ANSWER to REVIEWER 1

 Dear Referee,

Thanks for appreciating our work. Here are our answers to your comments:

General comments

1.    We can present the various food groups in the same way they appear in     Figure 1. It should be better but this would involve changing of numerical order of the bibliography with a consequent laborious work. We would prefer to leave the initial presentation if you agree. 
2.    In the â€śFIRST SECTION RESULTS” there are not sub headlines but only headlines. In the â€śFIRST SECTION DISCUSSION” there are headlines â€śVEGETABLE and FRUITS” (line 387), â€śDAIRY (MILK, CHEESE, YOGURT)” (line 418) and â€śRED MEAT and PROCESSED MEAT” (line 558) to facilitate the discussion since biological mechanisms are common to food subgroups. 
3.    We can try to be less wordy in the â€śFIRST SECTION RESULTS” but it is to note that the repetitions are often only apparent. For example the description of Table 11A is from line 307 to line 312. Instead from line 313 to line 321 we provide a detailed description of nonlinear dose-response meta-analyses. The Tables refer to linear dose-response meta-analyses only. In order to make this difference clearer, we changed the little title â€śDose-response analysis” to â€śNonlinear dose-response analyses” (line 96, 119, 152, 180, 199, 212, 222, 243, 253, 269, 279, 287, 298, 313, 330, 345). 
4.    We can try to be more accurate and concise in revising the â€śFIRST SECTION DISCUSSION”.a.    We can discuss biological mechanism of fruits and vegetables together and then delete lines 394-398 and lines 407-409. It will be: â€śThe protective mechanisms of increasing vegetable and fruit intake toward CVD risk include decreasing blood pressure, regulation of lipids metabolism, reducing oxidative stress and low-grade inflammation [81-83]. The high content of antioxidants (flavonoids, vitamin C, Vitamin E, Ăź-carotene) reduces DNA damaging [84]. 
 Specific comments
Main manuscript 
1.    Line 16: We can delete the first word â€śeffect” 
2.    Lines 17-19: We can clarify the sentence in lines 17-19 adding after (MDPPI) the sentence â€śThis pyramid comes from the Modern Mediterranean Diet Pyramid developed in 2009 for Italian people”. 
3.    Lines 33-34: In order to be more informative we can modify the sentence as following: â€śSome lifestyle modifications and among them the adoption of healthy dietary choices as well as increasing the intake of fresh fruits and vegetables, whole grains and healthy fats, represent useful measures for the prevention of NCDs [3].” 
4.    Lines 39-41: We don’t understand what kind of additional information would be useful to make the sentence more interesting. Actually, the sentence reported is the definition of dietary pattern according to DGAC. 
5.    Line 43: Of course we will change â€śGrece” to â€śGreece”. 
6.    Line 45: In order to be more accurate we can modify the sentence as follows â€śThe dietary habits characterised by higher consumption of vegetables and lower consumption of animal foods were considered important determinants of the very low mortality for CHD observed in the Corfu and Crete cohorts at 25-year of the follow-up [8]”. 
7.    Lines 47-48: We refer here to the dose-response meta-analyses that were object of our study. We identify 95 dose-response meta-analyses. In order to be more precise we can change the sentence in lines 47-48  to â€śDuring the last decade, a growing body of meta-analyses considered the relation between food groups and the development of NCDs”. 
8.    Line 51: In order to be clearer we can write â€ś to a variant of the Modern Mediterranean Diet Pyramid (www.inran.it) and the risk of CVD, including CHD and stroke, T2DM, CRC and breast cancer. This variant was the MDPPI [9] (Figure 1)”. 
9.    Lines 55-60: In order to be more accurate we can change the sentences as follows: â€śIn particular, we derived from these meta-analyses the serving sizes of food to be consumed in order to obtain a protective (Mediterranean food) or not detrimental (non-Mediterranean food) effect toward selected NCDs. A weekly menu plan was built on the advice about the frequency of intakes and the serving sizes of the MDPPI and the nutritional composition of this diet was compared with the Reference Italian Mediterranean diet followed in 1960 in Nicotera [10]. 

10.                Lines 129-134:  In order to be more clear we can rewrite as follows: “For stroke risk, there was no evidence of nonlinearity: risk reduction of stroke was evident along with the entire range of vegetable intake with the strongest reductions in RRs of 20% for a daily intake of 400 g [30]. However, in two dose-response meta-analyses there was evidence of nonlinearity between vegetable intake and stroke risk with a 28% reduction in RRs at 500 g/day [24] or with ~ 12% of risk reduction up to intakes of  ~ 200 g/day [17] with not additional benefits above these amounts.

11.                Line 549: we don’t understand why we have to rephrase this sentence.

12.                Line 556: we don’t understand why we have to rephrase this sentence.

13.                 Line 561: We refer here to nonlinear dose-response analyses   

  reported in lines 313-316.

14.            Lines 597-598: We don’t have any scientific basis to propose 1         serving equal to 25 g. We can modify the sentence in line 598 as                   follows: â€śWe propose arbitrarily that 1 serving is 25 g”                 

     15.         Line 659: In order to be more clear we can rewrite as follows: â€śIn 
                  the MDPPI the modification concerned cereal food which were 
                  divided into two groups”. 
     16.      We can change the little title â€śDose-response analysis” to â€ś 
                Nonlinear dose-response analyses” 
Line 90-112 Table 1. The lines 90-95 refer to a description of Table 1 (linear dose-response analyses). The lines 96-112 refer to â€śNonlinear dose-response analyses”. 
Lines 116-143Table 2A.  The lines 116- 118 refer to a description of Table 2A (linear dose-response analyses). The lines 119-143 refer to â€śNonlinear dose-response analyses”. 
Lines 147-171Table 2B.  The lines 147-151 refer to a description of Table 2B (linear dose-response analyses). The lines 152-171 refer to â€śNonlinear dose-response analyses”. 
Lines 175-190.  Table 3A. The lines 175-179 refer to a description of Table 3A (linear dose-response analyses). The lines 180-190 refer to â€śNonlinear dose-response analyses”. 
Lines 194-205. Table 3B. The lines 194-196 refer to a description of Table 3B (linear dose-response analyses). The lines 197-205 refer to â€śNonlinear dose-response analyses”. 
Lines 209-215. Table 3C. The lines 209-211 refer to a description of Table 3C (linear dose-response analyses). The lines 212-215 refer to â€śNonlinear dose-response analyses”. 
Lines 219-236. Table 4. The lines 219-221 refer to a description of Table 4 (linear dose-response analyses). The lines 222-236 refer to â€śNonlinear dose-response analyses”. 
Lines 240-245. Table 5. The lines 240-242 refer to a description of Table 5 (linear dose-response analyses). The lines 243-245 refer to â€śNonlinear dose-response analyses”. 
Lines 249-262. Table 6. The lines 249-252 refer to a description of Table 6 (linear dose-response meta-analyses). The lines 253-262 refer to â€śNonlinear dose-response meta-analyses”. 
Lines 266-272. Table 7. The lines 266-268 refer to a description of Table 7 (linear dose-response meta-analyses). The lines 269-272 refer to â€śNonlinear dose-response meta-analyses”. 
Lines 276-281. Table 8. The lines 276-278 refer to a description of Table 8 (linear dose-response analyses). The lines 279-281 refer to â€śNonlinear dose-response analyses”. 
Lines 285-291. Table 9. The lines 285-286 refer to a description of Table 9 (linear dose-response analyses). The lines 287-291 refer to â€śNonlinear dose-response analyses”. 
Lines 295-303. Table 10. The lines 295-297 refer to a description of Table 10 (linear dose-response analyses). The lines 298-303 refer to â€śNonlinear dose-response analyses”. 
Lines 307-321. Table 11A. The lines 307-312 refer to a description of Table 11A (linear dose-response analyses). The lines 313-321 refer to â€śNonlinear dose-response analyses”. 
Lines 325-338. Table 11B. The lines 325-329 refer to a description of Table 11B (linear dose-response analyses). The lines 330-338 refer to â€śNonlinear dose-response analyses”. 
Lines 342-348. Table 11C. The lines 342-344 refer to a description of Table 11C (linear dose-response analyses). The lines 345-348 refer to â€śNonlinear dose-response analyses”. 
Tables                                                           1. We checked ref 17. We found a                                                               mistake in the text in line 134 (~ 200                                                               and not ~400 g/day)                                                            2. Table 12. In our opinion Table 12 is a                                                                synthesis of our paper because it                                                                summarizes the frequency and the                                                                serving sizes of food groups. We                                                                would like to leave this table in the                                                                main text.                                                              3. Table 13. We planned a weekly                                                                  menu of about 2000 kcal/day and                                                                  we calculated the nutritional                                                                  composition of this diet on average                                                                  per day. 
Supplementary materials 
1. We agree with you. We will modify the weekly plan in order to consider â€śseasonality” and â€ślocal production”. 

Thank you very much for your help

Annunziata D’Alessandro, MD
 Reviewer 2 Report

The information in this manuscript is very useful and will be of greate interest to many readers as it deals with a nutrition-health issue of wide interest. The scope of the manuscript is impressive as it provides 126 very good references. The organization of the manuscript is also disciplined as it presents the information in terms of food categories.The figure 1 is very useful and is likely to be much reproduced once the manuscript is published. Also the writing style and English usage is very good. The tables indicating sources of information for foiod categories are very helpful. Discussion is particularly good in providing some general guidance regarding risk reduction  related to reasonable eating strategies. A very useful contribution is part 2 of the manuscript where the authors both provide recommended proportions of foods in table 12, and calorie analysis of the diet recommendatiuon in table 13.

However to make this manuscript better focused, there are several improvements that can me made.

1.       The authors provide the aim of the manuscript as summarizing the dose-response of health benefit for the quantity of category of food eaten (49-51):

“In the present study we performed a systematic review of dose-response meta-analyses of  prospective studies, which evaluated the association between the intake of food groups belonging  to a variant of MDPPI [9] (Figure 1) and the risk of CVD, including CHD and stroke, T2DM, CRC and breast cancer.

However, the authors do not deliver on this aim satisfactorily. For a true dose response result, the units of benefit should be stated in terms of increase of units of food. Instead, authors in most instances only give non-systematic statements (arbitrary amount of increment in benefit based on the increment in food intake derived from the data.) Rather than saying (116-118:” Each daily increment intake of 200 g of vegetables reduced the risk of CVD by 10% [24], whereas a daily increment intake of 400 g reduced the CHD risk by 18% [25] and a daily increment  intake of 100 g by 3% [17]. In these meta-analyses, the heterogeneity was below 50%.”, it would be more helpful and precise if they plotted  the two points of the CHD data to show a linear relationship  of y=0.05x-2. That way anyone with some math skills could then express how much lowering of CHD occurs for each unit increase of vegetable intake. The way authors express dose response may be intuitively OK, but it does not empower the reader with a more precise tool  of deciding how much of a given food to eat to achive what they individually need. Such information would be more explicit, than the vague discussion of the degree of non-linearity in the data. The reader does not benefit from such vague discussion because the authors do not reveal the details.

2.       There is a plethora of literature to the point of generating a general opinion, that the Mediterranean diet is beneficial. However, the diet is generally described in rather loose terms. In this manuscript as well. The authors state (42-45): “The Mediterranean Diet is a dietary pattern that was identified in the early 1960s in South Italy, Crete and other areas of Grece [6]. At that time the food intake habits of three cohorts of the Seven  Countries Study, Corfu and Crete in Greece and Nicotera in South Italy were almost identical [7]. Those dietary habits were considered important determinants of the very low mortality for CHD observed in the Corfu and Crete cohorts at 25-year of follow-up [8].”

This statement is not good enough. The reader of this review should not be forced to go to cited sources oin what people in Nicotera eat. We should be given a firm definition of Mediterranean diet (in Nicotera if that is used as a standard for this review), so that we know the proportions of foods and their macronutrients. The manuscript does not do this. And it is sorely needed. So  does Figure 1 represent a definition of Nicoteran ideal Mediterranean diet? If so, it should be clearly stated.

3.       Another apparent deficiency is a certain lack of specificity in describing food categories. Obviously the sources for this meta-analysis, many of which are meta analyses as well, probably do not provide enough specificity for the authors to be more specific.  However, reading about very interesting quantitative relationships between food categories and health benefits begs the question of specificity. The specific questions are, type of fats in some foods like milk, yovgurt and cheese. Are the data for full-fat milk products or milk products regardless of their fat content?(Skim milk is mentioned in discussion, line 421, but there is no general statement regarding the types of milk considered in the meta-analysis. That makes a difference because the health benefit of the fat content in foods in currently controversial. It makes a difference whether Nicoterans eat raw white cheese or processed cheddar cheese. While the authors may not have all the necessary information, it would be their responsibility to make an informed guess how the Nicoterans eat. Raw goat cheese (much lower in saturated fat than processed hard cheese). The same goes for other foods, Nuts, for instance differ depending whether they are tree nuts (where walnuts are known  for the highly beneficial levels of 3-omega fatty acids. What about peanuts? There is a general statement , lines 461-463, about “unique 461

 composition of these monounsaturated fatty acids and polyunsaturated fatty acids, fiber, 462

magnesium, arginine and polyphenols rich food [93]”, but why can’t the authors give a more specific rankings of nuts  considered in the meta-analysis, regarding these properties?Can we get a more nuanced sense of which nuts are good, and which are so-so?

Take the issue of fruit. Any fruit works as they present the data? What about watermelons and papayas/ Clearly, it is hard to create useful general data for food categories that differ so much in water and nutrient content as well as their sugar and phytonutrient content, but the authors should help the reader with some helpful interpretation based on the knowledge about which types of fruit were represented in source materials.

Or inclusion of fish and shellfish in the same category. Readers will want to know what type of fish? Sardines? Tuna. Salmon, eel?On lines 496-499, the authors state” the intake of 250 mg/day of eicosapentaenoic acid  and docosahexaenoic acid reduced CHD mortality by 36% without further reductions for higher intakes [101]. These intakes are easily obtained with two-100 g servings a week of which at least one is blu (should be “blue”)fish (www.ieo.it/bda).”, but the kind of fish rich in these fatty acids are not identified as they must have been in the data soiurce. What about cholesterol content of some shellfish? Again , the authors could calm the reader anxiety by providing some supplementary comments about vthe types of foods most frequently represented in data sources.

4.       I recommend that the authors do not use the term “bromatological” analysis for atble 13. First, most readers will not recognize the term. I had to look it up in Wikipedia. Second, the term is defined as “everything that involves the food consumed by living things. It studies each food and its components, from the moment it is produced, through its collection, transportation and even sale. It is a very specific work and involves different tasks, since Bromatology accompanies everything from the quality control of each food to its storage.” That is not what the table is for. It is actually “bioenergetics” or “caloric” analysis of food recommendation.

Author Response

ANSWER to REVIEWER 2

Dear Referee,

Thanks for appreciating our work. Here are our answers to your comments: 
1.     Our systematic review of dose-response meta-analyses concerned both linear and nonlinear dose-response analyses (59 meta-analyses). The results of these analyses (linear and nonlinear) were described in the  â€śRESULTS” of the â€śFIRST SECTION”. In the â€śDISCUSSION” of the  â€śFIRST SECTION” we made the effort to derive serving sizes of food groups to attain a protective (Mediterranean food) or a non-unfavourable (non-Mediterranean food) effect toward selected NCDs (CVD, including CHD and stroke, T2DM, colorectal cancer and breast cancer). We preferred in this study to define serving sizes considering more nonlinear dose-response analyses than linear dose-response analyses. The relation y= ax + b is applicable to linear regression only. It is very difficult to establish the RRs of a disease for a certain dose of food starting from several nonlinear dose-response analyses. However this could be the object of a new study. 
2. First, in order to be more accurate we can change (lines 45-46) â€śThose dietary habits were considered important determinants of the very low mortality for CHD observed in the Corfu and Crete cohorts at 25-year of the follow-up [8]” to â€śThe dietary habits characterised by higher consumption of vegetables and lower consumption of animal foods were considered important determinants of the very low mortality for CHD observed in the Corfu and Crete cohorts at 25-year of the follow-up [8]”. Second, we can write in the â€śDISCUSSION” of the â€śSECOND SECTION” the nutritional composition of the Nicotera diet in the late 1950’s reported by Prof. Fidanza. We will write â€śFidanza reported the following percentages of total energy of food groups: cereal (50-59%), virgin olive oil (13-17%), vegetables (2.2-3.6%), potatoes (2.3-4.4%), legumes (3-6%), fruit (2.6-3.6% including nuts that were about 3% of the weight of all fruit), fish (1.6-2.0%), red wine (1-6%), meat (2.6-5.0%), dairy (2-4%). The intake of eggs and animal fats was rare [95]”.Figure 1 is not exactly the Nicotera’s Mediterranean diet of the late 1950’s but in our opinion it is an excellent adaptation of this diet to our times. 
2.     Regarding the lack of specificity in describing food categories we completely agree with you. However, the epidemiological observational studies we reviewed usually do not give information about the type of fats in foods like milk, yogurt or cheese. However we know that in Nicotera’s diet of the late 1950s the dairy food were whole-fats and not processed. They were probably raw goat’s and sheep’s cheese that have a saturated lipid fatty acids profile less atherogenic than cow milk cheese. In our opinion what that is really important is that these food, if they are in small quantity as they were in the Mediterranean diet at that time, are not dangerous for health.In our study referring to nuts we mean tree nuts and peanuts. Providing detailed information on individual nut types is beyond the scope of our work.Also for fruit it is impossible to give specific information about the type of fruit in our systematic review of the dose-response meta-analyses. However, we will put this lack of specificity as one of the limits of our study in the conclusion of the paper.Regarding the fish and shellfish group the main result of our review is that a small intake of fish with a high content of n-3 long-chain PUFA (blue fish typical of Mediterranean Diet of Nicotera in 1960), can reduce the risk of CHD mortality. 
3.     We agree with you that it is better not to use the term â€śbromatological”. We will change this term in â€śnutritional”. 
 Thank you very much for your help

 Annunziata D’Alessandro, MD

 Round  2

Reviewer 1 Report

Dear authors,

I   would like to thank you for your reply, by I feel that some of the suggestions and the queries raised have not been taken into account.